# Epigenetics in Ovarian Cancer: A Review of Current Knowledge and Future Perspectives

**DOI:** 10.3390/biomedicines13112820

**Published:** 2025-11-19

**Authors:** Nikolaos Dedes, Michalis Liontos, Dimitrios Haidopoulos, Flora Zagouri, Kyveli Angelou, Anna Svarna, Athanasios Michas, Aikaterini Aravantinou Fatorou, Angeliki Andrikopoulou, Meletios-Athanasios Dimopoulos

**Affiliations:** 1Department of Clinical Therapeutics, Alexandra Hospital, Medical School, 11528 Athens, Greece; mliontos@gmail.com (M.L.); florazagouri@yahoo.co.uk (F.Z.); anna.svarna@hotmail.com (A.S.); athmixas@yahoo.gr (A.M.); k.aravantinou@hotmail.com (A.A.F.); aggelikiandrikop@gmail.com (A.A.); mdimop@med.uoa.gr (M.-A.D.); 21st Department of Obstetrics and Gynecology, Division of Gynecologic Oncology, Alexandra Hospital, National and Kapodistrian University of Athens, 11528 Athens, Greece; dimitrioshaidopoulos@gmail.com (D.H.); angeloukyveli@gmail.com (K.A.); 3Department of Medicine, Korea University, Seoul 04524, Republic of Korea

**Keywords:** ovarian cancer, epigenetic modification, chromatin alterations, histone modifications, novel clinical applications

## Abstract

Ovarian cancer is the gynecologic malignancy that bears the highest mortality rate in the Western world. This is attributed to late diagnosis and limited therapeutic progress. Recent advances in molecular oncology have highlighted the pivotal role of epigenetic modifications—including DNA methylation, histone modifications, non-coding RNAs, chromatin remodeling, and RNA methylation—in ovarian cancer development, progression, and treatment resistance. DNA methylation patterns affect key tumor suppressors and oncogenes, while histone modifications alter chromatin accessibility, influencing gene expression. Chromatin remodeling complexes, particularly the SWI/SNF complex, are frequently mutated in specific ovarian cancer subtypes, which is central in shaping their biological behavior. Non-coding RNAs, including microRNAs and long non-coding RNAs, further regulate tumor cell behavior and the immunosuppressive tumor microenvironment. Epigenetic profiles vary among histological subtypes and hold promise for biomarker development, early detection, prognosis, and therapeutic monitoring. Liquid biopsy approaches leveraging circulating tumor DNA methylation show diagnostic potential superior to conventional markers. Moreover, targeting epigenetic regulators—such as DNMT and HDAC inhibitors, EZH2 antagonists, and RNA-modifying enzymes—offers novel avenues for treatment, particularly in reversing chemoresistance and sensitizing tumors to immunotherapy. While promising, these strategies require further validation through clinical research to translate into effective clinical interventions. This review aims to summarize the current literature and highlights potential applications of epigenetic manipulation in day-to-day practice.

## 1. Introduction

Ovarian cancer is a major public health issue worldwide and currently stands as the most lethal gynecologic malignancy in Europe and North America. In the absence of any meaningful screening strategy, the high mortality rate of this disease can be ascribed to the fact that most patients are diagnosed at advanced stages when symptoms develop [1]. Furthermore, there has been relatively little progress in the development of more effective treatments, as for decades the main treatment axis comprises optimal cytoreduction and cytotoxic chemotherapy.

Nevertheless, key to achieving better patient outcomes is the thorough study of the disease’s pathophysiologic basis. In this context, ovarian tumors are divided into groups according to the cell origin. Therefore, they may be of epithelial, stromal, or germ cell origin [2]. Epithelial ovarian carcinomas make up the vast majority (up to 90%) of ovarian malignancies. This large group is further subdivided into high grade serous ovarian carcinomas (HGSOC), low-grade serous ovarian carcinomas (LGSOC), endometrioid carcinomas (EC), and clear-cell carcinomas (CCC). Both HGSOC and LGSOC originate from the fibrial or ovarian surface epithelium, but their mutational status is quite different as the former is characterized by p53 and BRCA mutations, while the latter harbors mutations of KRAS and BRAF. Moreover, EC and CCC may be associated with endometriosis and mutations of PTEN and ARID1. Furthermore, 6% of ovarian tumors are of mesenchymal origin, and the remainder comprises the germ cell-derived small cell carcinomas, of which the hypercalcemic type (SCCOHT) is of particular interest. This histology afflicts young women, has aggressive biologic behavior, and is defined by SMARCA2/4 mutations [3].

Scrutiny of the underlying molecular mechanisms of the disease is of paramount importance for the optimization of cancer treatment. Relative to this, recent advances of our understanding of DNA repair mechanisms have yielded considerable results. The prime example is the use of poly ADP-Ribose polymerase inhibitors in patients with HGSOC, and deficiency in the homologous recombination repair mechanism offers clinical benefit. The exploitation of certain genetic traits is of unequivocal value, yet further gains may be achieved through the elucidation of epigenetic modulation. In spite of the paucity of the available literature, epigenetic changes seem to play a pivotal role in ovarian cancer’s natural history. As such, ovarian cancer epigenetics may be the basis for novelties in disease management. 

This review endeavors to highlight the role of epigenetics in all aspects of ovarian cancer. The initial aim is to investigate epigenetic modification in disease natural history and its pathophysiology. Knowledge of the underlying molecular processes may help develop practice-changing clinical applications in disease screening and diagnosis. Moreover, epigenetic manipulation via a series of therapeutic agents has already yielded results in other fields of medicine. In this context, there is sufficient evidence to speculate that such drugs may be efficacious in ovarian cancer as well. As such, this review may offer insight into future perspectives of ovarian cancer management.

## 2. Epigenetics in Ovarian Cancer Development

Scientific work of the last two decades has revealed that differential DNA-methylation, histone modifications and non-coding RNAs (ncRNAs) may be involved in ovarian cancer development. Some of the most prominent genes or pathways subjected to epigenetic modification during ovarian cancer pathogenesis are listed in Table 1.

A number of hotspot genes have been reported to be inactive due to promoter hypermethylation. DNA methylation is regulated by a group of three DNA methyltransferases (DNMT). BRCA1/2 is one of many DNA repair genes that are epigenetically deactivated in ovarian cancer [4]. Furthermore, cell cycle regulatory genes such as the cyclin-dependent kinases may be similarly “switched-off”. On the contrary, hypomethylation of proto-oncogenes is a documented event in ovarian cancer. Hypomethylation of oncogenes such as BORIS or imprinted IGF-2 may be involved in malignant transformation [5,6]. Moreover, promoter hypomethylation in HOXA10 in HGSOC is associated with decreased survival [7]. It should be noted, however, that the methylation-profile may be altered along the disease course, as satellite DNA and some genes such as LINE-1 and CT-45 are preferentially hypomethylated late in ovarian cancer history [8,9,10]. Finally, there seems to be a distinct methylation profile between ovarian cancer subtypes. HGSOC has a methylation profile distinct from LGSOC, while HGSOC also displays lower levels of methylation than CCC and EC [11].

Next to DNA methylation, histone modifications (e.g., acetylation, phosphorylation, methylation, ubiquitination) are of pivotal importance to selective DNA expression as these are key regulators of chromatin structure. Of the various histone modifications, histone acetylation (promotes euchromatin—i.e., chromatin in relaxed state) and histone methylation (promotes eterochromatin—i.e., condensed chromatin state) are the most studied. The attachment of acetyl groups to the N-terminus is termed “acetylation” and is mediated by the histone acetelylases (HATs), while the removal of the acetyl groups is termed “deacetylation” and is achieved through the activity of the histone deacetylases (HDACs) [12]. In normal ovarian tissue the activities of these opposing groups are in balance, the disturbance of which is main feature in ovarian cancer [13]. In this context, human MOF, a member of the HATs, has been implicated in ovarian cancer development and prognosis [14]. In comparison to the HATs, the HDACs have received far more scientific attention. There are 4 groups of HDACs, namely HDAC I, II, and IV are zinc-dependent enzymes, while HDAC-III is NAD+ dependent [15]. HDAC-I has been implicated in cancer development and progression partly through its ability to inhibit the RGS-2 gene, whose protein normally acts in a tumor-suppressing manner [13]. Interestingly, HDACs-I influence tumor cell migration through the concurrent promotion of Cyclin A and inhibition of E-cadherin, while there seems to be a correlation between HDAC levels in tissue ki67 expression [16]. Moreover, HDAC-II members may influence key proteins such as HSP90, MSH-2, and p53, while they also enhance the signaling of pathways such as RAS, PI3K/Akt, and MAPK [17,18,19,20]. One particular member, HDAC9, is involved in epithelial to mesenchymal transition (EMT) with an inverse impact on prognosis [21]. In addition to these, sirtuin-1—a HDAC-III—may be directly connected to ovarian cancer development and progression, as it obstructs DNA repair through p53 deacetylation and promotes platinum resistance [22,23], both of which are found to impact patient prognosis. Nevertheless, histones are not the sole substrate of HDACs [24]. In ovarian cancer HDACs also participate in disease progression via the remodeling of the cytoskeleton, the nuclear translocation of b-catenin, and the induction hypoxia inducible factor 1a (HIF-1a) [25]. The upregulation of HIF-1a may be crucial in HSOC natural history, as it seems to be responsible for the upregulation of Snail and the inhibition of E-cadherin [26]. In addition to these, HIF-1a also targets GLUT-1, VEGF, CXCR-4, OCT-4, and SOX-2. All these genes seem to participate in EMT and are central to glucose metabolism, angiogenesis, inflammation, migration, and cell stemness [27]. In parallel to histone acetylation, histone methylation regulation is mediated through the activity of histone methyltransferases (HMTs) and demethylases (HDMs). Of the HMTs, particular mention is reserved for the Enhancer of Zeste Homolog 2 (EZH2)—a component of the polycomb repressive complex (PRC) [28]. This enzyme may contribute to tumorigenesis through the epigenetic suppression of tumor-suppressor genes [29]. Meanwhile, EZH-2 mediates malignant transformation via regulation of TGF-beta and cell migration due to downregulation of E-cadherin [30]. Further disease progression is aided by EZH-2 mediated H3K27 triple methylation, which drives cell proliferation, fibroblast recruitment, angiogenesis, and metastasis development [31]. Increased EZH-2 activity can be found in up to 85% of epithelial ovarian cancer cases, yet its biological impact is influenced by the interplay of EZH-2 and other epigenetic regulators such as the components of the BAF complex [32,33]. Loss of EZH-2 antagonists such as ARID1A and SMARCA2/4 leads to the repression of tumor suppressor genes in ovarian cancer [34,35,36,37]. Another notable HMT is arginine methyltransferase 5. Its control over E2F-1, a cell cycle regulator, promotes tumorigenesis through increased cell proliferation [38]. Aside from HMTs, HDMs may also contribute to ovarian carcinogenesis. Lysine-specific histone demethylase 1 and the KDM subfamily of HDMs are associated with cell proliferation, p53 targeting, and migration [39,40].

Both DNA methylation and histone alterations are the most recognized mechanisms of epigenetic modification. Nonetheless, in recent years ncRNAs have been increasingly recognized for their role in epigenetics. Among those, the most memorable are the microRNAs (miRs), which may regulate translation and post-translational processes and the long non-coding RNAs (lncRNAs), which influence chromatin density [41,42]. Review of recent literature has shown that different miRs may exert pro- or anti-neoplastic effects. As such, elevated levels of miR-9 may mediate tumorigenesis, while miR-205 drives migration via downregulation of ZEB1 [43,44]. To add to these, miR-96 and miR-182 are reported to inhibit the tumor-suppressing Forkhead box O3 [45]. However, miR-101 exhibits anti-tumor activity by targeting EZH-2 [46]. Similarly, miR-34 suppresses tumor cells, and it is usually downregulated in cancer [47]. Other notable mentions of tumor-suppressing miRs include miR-200 (increases expression of E-cadherin), miR-214 (blocks the PTEN/Akt pathway), and miR-203 (Snail-2 silencing—inhibition of EMT) [48,49,50]. Relative to miRs, lncRNAs have received less attention, yet there are mentions of lncRNAs that may promote tumor progression either through signaling via the Notch or the FAK/PI3K/Akt pathways or the overexpression of metalloproteinases—thus facilitating tumor invasion [51,52]. Finally, there are some sporadic mentions of anti-tumor lncRNAs [53].

**Table 1 biomedicines-13-02820-t001:** Major genes and molecular pathways subjected to epigenetic modification in ovarian cancer pathogenesis.

Gene or Pathway	Epigenetic Modification and Its Impact
BRCA1	Increased methylation in individuals with familial ovarian cancer compared to sporadic cancer [54] Increased methylation is associated with higher probability of PARP inhibitor efficacy [55] Methylation of Fanconi-anemia gene, which may interact with BRCA leads to cisplatin resistance [56] Hypermethylation of BRCA1 is more frequent in serous than endometrioid ovarian cancer [57] Bilateral ovarian cancer displays higher levels of BRCA1 methylation [58,59]
p53	Uniformly methylated in malignant ovarian tissue [60] Hypermethylation of YPEL-3 -a tumor suppressor downstream of p53 linked to cellular senescence- may be associated with disease progression [61]
RASSF1A	RASSF1A methylation is increased in ovarian cancerous tissue compared to normal tissue or benign lesions [62,63] Methylation status of RASSF1 may predict response to treatment and survival in the neoadjuvant setting [64]
CHD5, FBP1, ALDH1A2, FOXD3, IGFBP-3, ZNF671, SPARC	Notable tumor-suppressor genes, which are hypermethylated in ovarian cancer [65]
HOXA9	Differentially methylated in HGSOC and benign lesions [66] Lower grades of HOXA9 methylation are associated with higher tumor grade [67] Methylation status of HOXA9 may influence response to PARPi treatment [68]
CBX8	Hypomethylation of CBX8 may serve as prognostic biomarker [69]
SLC6A12, AGR2, GABRP	Notable oncogenes commonly hypomethylated in ovarian cancer [65]
Wnt/β-catenin pathway	Wnt/β-catenin pathway may be upregulated in ovarian cancer due to SFRP methylation, which functions as a pathway inhibitor. SFRP methylation is associated with worse prognosis, aggressive biological behavior and increased risk of recurrence [70] IQGAP2 and THEM88 (both0 inhibitors of Wnt/β-catenin signaling) may be hypermethylated in ovarian cancer—a feature associated with disease progression [71,72]
TGF-β pathway	Epigenetic silencing of FBXO32, ABCA1, SOX2 and TGFBI is associated with disease progression due to enhanced TGF-β signaling [65]

## 3. The Influence of Chromatin Remodeling on Different Histologies of Ovarian Cancer

Chromatin configuration is perhaps the most important control of gene expression, as DNA needs to be accessible to the transcription mechanism. As such, nucleosome density is primarily regulated by nucleosome remodelers like the ATP-dependent chromatin remodeling complexes Chromodomain helicase DNA binding (CHD), Inositol requiring 80 (INO80), Imitation switch (ISWI) and Switch/sucrose non-fermentable (SWI/SNF, also referred to as BAF).

The fundamental function of this molecular machinery is the mobilization and organization of nucleosomes so as to control access to genetic information [73].

Of the chromatin remodeling complexes, BAF is the most studied. BAF is mutated in 20% of human cancers and major components of this enzyme complex like ARID1A and SMARCA2/4 are consistently mutated in specific histologies of ovarian cancer [74]. ARID1A may be mutated in 57% of CCCs and 30% of EC [3,75]. These mutations usually lead to loss of function and the resulting change in chromatin configuration does not allow for the transcription of CDKN1A and PIK3IP1, both of which lead to apoptosis [35,36,75]. Furthermore, another interesting observation is that an antagonistic behavior exists between the BAF and PRC, as well as between the BAF and HDACs [76]. Therefore, EZH inhibition or HDAC inhibitor therapy may be a rational basis for treatment of patients with CCC or EC [77]. Besides ARID1A, the other notable BAF components are SMARCA2 and SMARCA4. These genes may be mutated in lung cancer, melanoma, and lymphoma, yet loss of this protein is the hallmark of SCCOHT [78,79]. This phenomenon seems to be the pathophysiologic base of this rare form of ovarian cancer, as EZH-2 inhibition seems to have therapeutic potential. Furthermore, another important interaction is that between the BAF and coactivator-associated arginine methyltransferase 1 (CARM1) [80]. CARM-1 takes part in major subcellular processes and is frequently overexpressed in HGSOC. CARM-1 may methylate SMARCC1 (a BAF component), which may displace the BAF from its locus [81]. In a similar manner, CARM-1 and EZH2 compete for the control of key tumor suppressor genes [82]. Therefore, EZH-2 targeting may be a feasible target in HGSOC treatment.

## 4. RNA Modification in Ovarian Cancer

RNA modification may not be included in the usual context of epigenetics, as it is an emerging but no less exciting field. Accumulating evidence suggests that regulation of RNA metabolism may be key to ovarian cancer pathophysiology. A multitude of RNA modifications have been described, yet the most studied revolve around the methylation and demethylation of RNA at various sites (e.g., methylation of adenosine at the 6th position—m6A)—a process that drastically alters their biological impact. In parallel with DNA epigenetics, the enzymes that mediate RNA methylation are referred to as “writers”, while those who remove methyl-groups are termed “erasers” [83]. As such, these processes may influence RNA processing, alternative splicing, stability, kinetics, and translation. In this context, “readers” such as METTL3 and VIRMA have been associated with apoptosis inhibition, migration, and invasion in ovarian cancer, while FTO leads to reduced “stemness” in ovarian cancer cell lines and YBX1 renders malignant cells susceptible to the cytotoxic effects of cisplatin [84,85,86,87]. Despite not having received the same attention as epigenetics, “epitrascriptomics” may help to further clarify the molecular basis of ovarian cancer, as the mapping of RNA modifications transcriptome-wide could enhance our understanding of its pathogenesis. In this context, novel biomarkers may be developed, and manipulation of epigenetic alterations could lead to new treatment options.

## 5. Epigenetic Modifications in Ovarian Cancer Tumor Microenvironment

The tumor microenvironment (TME) refers to immediate surroundings of the malignant cells, and it includes the tumor stroma, immune cells, endothelial cells and fibroblasts. As may be expected, epigenetic modification is central to the formation and the dynamics of the TME. Modern understanding of cancer biology has highlighted the TME as crucial to disease physical history and response to treatment. Mounting evidence suggests that TME characteristics actively direct disease course 

Evidence from studies in various malignancies demonstrates that stromal fibroblasts can acquire DNA methylation profiles similar to adjacent malignant cells, indicating reciprocal epigenetic crosstalk between the stroma and tumor compartments. In this context, cancer-associated fibroblasts (CAFs) secrete soluble factors that modulate the methylation status of epithelial ovarian cancer cells, promoting widespread transcriptional changes, EMT, and acquisition of stem-like features—effects that can be amenable to therapeutic targeting. A key mediator of this interaction is TGF-β1, which is released by fibroblasts in response to tumor-derived signals. In ovarian cancer, TGF-β not only induces global DNA hypermethylation and EMT through upregulation of DNA methylation but also contributes to the reprogramming of fibroblasts into pro-tumorigenic CAFs that secrete prometastatic mediators. This establishes a self-sustaining feedback loop wherein fibroblast–tumor interactions perpetuate epigenetic remodeling, CAF activation, and tumor invasion. Targeting this loop through the combined use of demethylating agents and TGF-β pathway inhibitors therefore represents a promising therapeutic avenue to disrupt the epigenetic reinforcement of the ovarian cancer microenvironment [88]. Moreover, DNA methylation plays a pivotal role in shaping the immunosuppressive landscape, particularly through its impact on myeloid-derived suppressor cells (MDSCs). These heterogeneous immune cells expand in response to malignant transformation and inflammation, where they suppress anti-tumor T-cell activity and promote tumor progression. Elevated MDSC levels in ovarian cancer patients are strongly correlated with reduced overall survival, highlighting their clinical relevance. Pro-tumorigenic cytokines such as interleukin-6 (IL-6) and interleukin-10 (IL-10) enhance MDSC recruitment and function through activation of the STAT3 signaling pathway, while VEGF and adenosine secreted by ovarian cancer cells further facilitate their accumulation and immune-suppressive effects. Importantly, recent evidence indicates that MDSCs undergo DNA methyltransferase 3A- and prostaglandin E2-dependent hypermethylation, a process essential for their acquisition of immunosuppressive properties. This epigenetic reprogramming not only reinforces immune evasion within the TME but also underscores the potential of targeting methylation pathways as a novel therapeutic strategy to restore anti-tumor immunity in ovarian cancer [89,90].

In addition to DNA methylation, histone modifications within the ovarian cancer TME have emerged as key regulators of tumor progression, metastasis, and therapeutic response by influencing both stromal and immune cell behavior. Beyond cancer cells, epigenetic alterations—including histone methylation and acetylation—profoundly shape the phenotype of stromal components such as fibroblasts, adipocytes, and immune cells. Notably, nicotinamide N-methyltransferase (NNMT) was found to be upregulated in CAFs within metastatic ovarian cancer lesions, where it depletes S-adenosyl methionine, the principal methyl donor for histone methylation. This depletion leads to reduced global histone methylation, promoting transcriptional reprogramming that enhances collagen secretion, cytokine release, and metastatic potential [91]. Conversely, inhibition of NNMT restores histone methylation and suppresses expression of metastasis-associated genes, underscoring its central role in stromal remodeling. Platinum-based chemotherapy further alters histone dynamics in the TME by inducing IL-6 secretion from CAFs, which drives stemness-associated pathways in residual tumor cells. Additionally, epigenetic silencing of immune-related cytokines through DNA and H3K27 methylation limits immune infiltration, contributing to the “cold” immune phenotype characteristic of OC. Restoration of histone methylation balance using hypomethylating or histone methyltransferase inhibiting agents can reactivate immune signaling pathways, including CCL5–CXCL9/CXCL10 axes, thereby enhancing T-cell infiltration and responsiveness to immune checkpoint blockade [92]. Furthermore, cytokines and signaling molecules such as IL-33, IL-6, GM-CSF, IL-4, and IL-10 orchestrate histone modification patterns in tumor-infiltrating MDSCs and tumor-associated macrophages (TAMs), enhancing their immunosuppressive functions. For example, IL-33 promotes survival of MDSCs by increasing activating histone marks like H3K4me3 and H3K14ac, while IL-6 and GM-CSF induce STAT3- and C/EBPβ-dependent transcriptional programs that regulate H3K4me3 enrichment at promoters of immunosuppressive genes such as Arg1, NOS2, and COX2. Similarly, IL-4-driven STAT6 signaling upregulates JMJD3, a histone demethylase that removes repressive H3K27me3 marks from TAM-related genes, sustaining the tumor-promoting macrophage phenotype. IL-10 also contributes to this suppressive milieu by recruiting histone demethylases such as KDM6A to remove H3K27me3 and promote transcription of Hotairm1, facilitating nuclear localization of S100A9 and expansion of MDSCs. Collectively, these findings highlight histone modification as a central epigenetic mechanism coordinating tumor–stroma–immune interactions in ovarian cancer and suggest that targeting histone-modifying enzymes offers a promising strategy to reprogram the TME and improve therapeutic outcomes [93,94].

Beyond DNA methylation and histone modifications, ncRNAsare emerging as key instruments of epigenetic and post-translational modification. Non-coding RNAs, including microRNAs (miRNAs), long non-coding RNAs (lncRNAs), and circular RNAs (circRNAs), play crucial regulatory roles in the TME, influencing cancer progression, metastasis, immune evasion, and therapy resistance [94]. Genetic alterations—both germline and somatic—can occur within ncRNA genes, affecting their expression and function. For example, differential miRNA expression patterns have been explored as diagnostic biomarkers for hereditary breast and ovarian cancers, while somatic mutations in miRNA genes and their target sites (such as in 3′UTRs) can disrupt regulatory interactions and promote oncogenesis [95]. Within the TME, communication among tumor, stromal, and immune cells is mediated by cytokines and ncRNAs. Cytokines such as IL-6, IL-10, and IFN-γ can modulate ncRNA expression, which in turn regulates pathways like PI3K/Akt, NF-κB, and JAK/STAT [96]. For instance, lncRNA ZFAS1 promotes cancer progression by sponging miRNA-6499-3p and upregulating CCL5, while lncRNA INCR1 enhances PD-L1 expression through IFN-γ signaling. Exosomes act as key ncRNA carriers within the TME, transporting miRNAs, lncRNAs, and circRNAs that regulate tumor growth, angiogenesis, immune suppression, and drug resistance. Exosomal ncRNAs such as miRNA-1246 and miRNA-6126 modulate pathways involved in ovarian cancer progression and therapy response [97]. These vesicles also influence immune cells—promoting macrophage polarization, impairing T-cell function, and altering dendritic cell activity—thus shaping immune evasion. Cancer-associated fibroblasts, activated by growth factors and hypoxia, express ncRNAs that remodel the extracellular matrix and enhance metastasis. In ovarian cancer, CAF-derived CXCL14 increases lncRNA LINC00092, promoting invasion [98]. Similarly, TAMs, primarily of the M2 phenotype, rely on ncRNAs for recruitment and polarization; for example, miRNA-149 suppresses TAM infiltration, while siRNAs targeting VEGF and PIGF reprogram TAMs to an antitumor M1 state [99]. Non-coding RNAs also regulate T-cell and B-cell functions in the TME. Modulation of immune checkpoints (PD-1/PD-L1) via siRNAs can restore T-cell cytotoxicity, while ZFP91 knockdown enhances T-cell metabolism and activity [100]. B-cell activity is influenced by ncRNAs downstream of CXCL13/CXCR5 signaling, affecting proliferation and migration. Finally, ncRNAs govern EMT, a key process in metastasis and drug resistance. Downregulation of miRNA-200b/c promotes EMT and tamoxifen resistance, while restoring their expression reverses these effects [101]. Collectively, ncRNAs serve as both mediators and messengers within the TME—regulating cytokine signaling, intercellular communication, immune responses, and treatment sensitivity—making them promising diagnostic biomarkers and therapeutic targets in cancer. 

With regard to the far-reaching influence of epigenetics on the TME, it would be rational to consider the therapeutic potential of epigenetic modulation with the aim to reprogram immune and stromal components of the ovarian cancer TME toward antitumor activity. DNA methylation inhibitors, such as 5-azacytidine (5-aza) and 5-aza-2′-deoxycytidine (5-aza-dC), can reprogram immunosuppressive M2 TAMs into proinflammatory M1 phenotypes, enhancing tumor sensitivity to chemotherapy such as paclitaxel [102]. Moreover, combination therapies involving DNA methylation modulating agents and histone deacetylase inhibiting agents (e.g., trichostatin A, TSA) further remodel the cytokine milieu, promoting antitumor immune responses [103]. In regulatory T cells (Tregs), modulation of FOXP3 methylation alters their suppressive capacity, with hypomethylating agents transiently increasing FOXP3 expression but ultimately reducing Treg-mediated immunosuppression, thereby enhancing checkpoint inhibitor efficacy [103]. Histone modifications also represent a promising therapeutic avenue. Selective inhibition of specific HDACs (e.g., HDAC6 by ACY-1215, HDAC3 by RGFP966) enhances CD8^+^ T-cell cytotoxicity while limiting Treg function, whereas inhibitors of EZH2 (UNC1999, EPZ005687, CPI-1205) promote NK cell activation and facilitate conversion of Tregs to Th1-like cells, boosting IFNγ production [104]. Targeting acetyltransferases (EP300/CBP inhibitors) similarly disrupts FOXP3 acetylation, reducing Treg stability and improving immune activation within the OC TME. RNA modification inhibitors have recently emerged as an additional layer of epigenetic intervention. Inhibitors of m^6^A regulators such as FTO (e.g., Dac51) and ALKBH5 (ALK-04) suppress the infiltration of immunosuppressive MDSCs and Tregs, enhancing anti-PD-1 therapy efficacy [105]. Similarly, METTL3 inhibitors (STM2457) and pseudouridine synthase (PUS7) inhibitors (C17) demonstrate antitumor activity and improved survival in preclinical models [106]. Collectively, these findings underscore the therapeutic promise of combining epigenetic drugs with immunotherapies to reprogram the ovarian cancer TME. By targeting DNA methylation, histone modification, and RNA modification pathways, it may be possible to overcome immune resistance, restore effector cell activity, and enhance the efficacy of immune checkpoint blockade in ovarian cancer.

## 6. Epigenetic Modification as a Means of Disease Diagnosis and Surveillance

It is an established fact that early diagnosis is key to achieving better survival, yet the only available and most practical biomarker, cancer antigen 125, offers limited specificity and sensitivity. Therefore, there is an unmet need for novel biomarkers. 

As such, the detection of DNA methylation in circulating tumor DNA may be a candidate biomarker for the timely detection of ovarian cancer [107]. In this context, the methylation profile of key tumor-suppressor genes has drawn attention due to its high diagnostic accuracy. Relative to this, specificity and sensitivity may be further increased by inquiring for a set of hotspot genes. Furthermore, the evaluation of DNA-methylation in cell-free DNA seems to be superior to cancer antigen 125 (CA125) as a tool of evaluation of ovarian cancer risk and in discerning healthy women from ovarian cancer patients [108]. Besides disease diagnosis, DNA methylation may also be implemented in the setting of ovarian cancer monitoring. As such, methylation analysis of genes like HOXA9 and HIC1 in conjunction with cancer antigen 125 levels are reported to offer efficient and effective surveillance of ovarian cancer [109,110]. 

Histone modifications have also been the subject of scrutiny as to their possible function as a diagnostic biomarker. As such, both hMOF (a member of the HATs) and bromodomain-containing proteins (proteins that have the ability to bind to acetylated lysine residues) have been tested as diagnostic tools [111,112]. Furthermore, EZH-2 mRNA levels have been postulated to be an effective predictive biomarker, as this protein is known for its ability to universally trimethylate H3K27, which in turn is strongly connected to disease progression [31]. Moreover, two other members of the HATs have shown promise in the setting of disease diagnosis. SETD8 levels appear to be higher in HGSOC than in normal tissue without further correlation with disease stage, while dpy-30 homolog protein has been linked with advanced stage and patient survival in epithelial ovarian cancer [113,114].

In recent years, liquid biopsy has gained considerable ground as a means for non-invasive diagnosis and surveillance. The methylation of cfDNA highly correlates tumor DNA methylation profile [115]. As such, the study of cell free DNA (cfDNA) methylation profiles has come to the center of attention as an accurate and practical disease biomarker [107]. Most of the available literature on this subject is derived from studies on breast, colon, liver, and lung cancer—yet there is sufficient evidence suggestive of its utility in ovarian cancer as well. The initial data has been derived from single gene methylation biomarkers such as BRCA1 and RASSF1. There seems to be considerable concordance (>80%) in the methylation profile of these two genes between serum and tissue samples. Furthermore, cfDNA methylation of BRCA1 and RASSF1 was not found to be elevated in patients with benign ovarian histologies [116,117,118]. In a similar fashion, OPCML methylation may be useful in detecting ovarian cancer. The frequency of methylation may correlate with disease stage, as it was found to be 42.9%, 66.7%, 85.7%, and 100% for FIGO stage I, II, III, and IV, respectively [119,120]. In another study, OPCML methylation was able to detect early ovarian cancer with sensitivity and specificity around 90% [119]. Similar accuracy in the detection of early ovarian cancer has been achieved by the evaluation of HOXA9 methylation [120]. In an effort to increase the performance of cfDNA methylation assays, many researchers sought to concurrently test for several hotspot genes (e.g., BRCA1, RASSF1, CDH1, OPCML, etc). Such testing panels appear to achieve specificity and sensitivity of up to 90% in detecting early ovarian cancer (stages I and II). In addition to this, they seem to be able to discern ovarian cancer from borderline malignant tumors and benign pathology. In spite of its possible applications in disease diagnosis, cfDNA methylation tests may also be utilized in the setting of disease and treatment monitoring. In general, cfDNA may serve as a surrogate of disease burden and a marker of disease recurrence post optimal cytoreduction or chemotherapy [121,122,123,124]. However, detection of specific epigenetic modifications detected through liquid biopsy may be used to guide treatment decisions. Acquired (post-paclitaxel/carboplatin) hMLH1 methylation may be a marker of treatment resistance and a predictor of survival [125], while the methylation status of SLFN11 was associated with progression-free survival (PFS) in patients with HGSOC [125]. Meanwhile, cfDNA methylation seems to be altered by treatment administration. HOXA9 promoter methylation was evaluated in patients before and after 3 cycles of chemotherapy. The patients who still had detectable levels of cfDNA after 3 rounds of treatment had significantly lower PFS and overall survival (OS) than patients without detectable cfDNA [68]. In another study, a set of three DNA methylation markers measured before and after two cycles of treatment was able to better define responders from non-responders than the cancer antigen 125 cut-off of 35 units/milliliter [126]. Further studies postulate the use of RASSF1A methylation as a disease surveillance measure, while some experts have even suggested the assessment of cfDNA in ascites [127].

## 7. Therapeutic Applications of Epigenetic Modifications

To date, ovarian cancer is treated through a combination of cytoreductive surgery and chemotherapy. Although most patients initially respond well to therapy, the vast majority will have disease recurrence within three years of diagnosis. As a result, 5-year survival rates are rather disappointing in spite of the introduction of novel treatments such as PARP inhibitors and immunotherapy. This is mainly attributed to the emergence of resistance to therapy that is partially due to underlying epigenetic changes. Therefore, attempts to reverse treatment resistance may lead to improved efficacy of available treatments [128].

Among many epigenetic targets, use of DNMT inhibitors (DNMTi) has been for long the subject of scientific research. DNMTis include analogs of deoxycytosine, which are potent inhibitors of methyl transfer [129]. Among those, decitabine and 5-azacytidine have proven their efficacy in the treatment of hematological malignancies and myelodysplastic syndromes [128]. As such, the FDA has approved their use due to their verified anti-tumor activity via the inhibition of DNA methylation. DNMTis have also been tested in the treatment of chemotherapy-resistant ovarian cancer with the aim of restoring platinum sensitivity [130]. In this context, patients were treated with low-dose decitabine for five days prior to platinum administration. This regimen was linked to increased demethylation and re-established carboplatin sensitivity. The molecular mechanisms leading to the reinstitution of platinum sensitivity are not elucidated—yet this may be the function of altered TGF-b signaling and the reduced methylation of genes like HOXA10, HOXA11, BRCA1, MLH1, and RASSF1a [130]. Phase I and II trials of DNMTis in combination with platinum mention overall response rates up to 35% with reasonable PFS intervals [130,131,132,133,134,135]. Another similar agent, 5-azocytidine, may have comparable functions as decitabine with less efficacy in restoring platinum sensitivity. A major issue with DNMTis is myelotoxicity, which may cause dose-limiting adverse reactions (e.g., neutropenia). This matter could be partially addressed with the advent of second generation DNMTis such as guadecitabine [136].

In parallel to DNMTs, HDAC inhibitors (HDACi) may be effective in restoring transcriptional alterations. As is the case with DNMTis, vorinostat, belinostat, panobinostat, and romidepsin are FDA-approved HDACis for the treatment of hematologic malignancies [137]. There is also considerable evidence to support HDACi efficacy in ovarian cancer as well [25,138,139]. Histologies harboring ARID1A mutations (e.g., CCC or EC) may be susceptible to the pan-HDAC inhibitor SAHA via the suppression of HDAC2 or HDAC6 [77]. The latter deacetylates Lys120 on p53 and may influence entry into apoptosis. Furthermore, belinostat may be coupled with decitabine in an effort to re-establish platinum sensitivity in platinum-resistant ovarian cancer [140]. Moreover, special mention is reserved for the rare histology of SCCOHT, whose pathophysiology seems to be causatively linked to biallelic loss of SMARCA4. Hsu et al. [141] report a remarkable case of a young patient with SCCOHT, who after multiple cytoreductive operations and consecutive lines of intensive chemotherapy was treated with tazemetostat, a selective inhibitor of histone methyltransferase EZH-2. The patient achieved a remarkable remission that lasted four years. Treatment with the EZH-2 inhibitor was discontinued due to the appearance of T-cell acute lymphoblastic leukemia, which is possibly linked to the long-standing inhibition of EZH-2 [141]. Clinical trials of DNMTis and HDACis in chemoresistant ovarian cancer are shortly displayed on Table 2.

Another point of interest is the manipulation on non-coding RNAs, as preclinical data so promise for the development of new therapeutic applications. In recent years, a multitude of miRs have been proposed as candidate targets. One such is miR-181a, whose inhibition has been suggested to be of value as it is an activator of the Wnt/b-catenin pathway, which in turn has been linked to disease progression in HGSOC [144,145]. Similarly, miR-200c and miR-9 have been associated with PARP inhibitor efficacy [145,146]. Leveraging lncRNAs is another novel approach. One such example is HOTAIR, inhibition of which has been suggested as a means of overcoming carboplatin resistance via the downregulation of HOXA7 and Wnt/b-catenin pathway [147,148].

Finally, there is the combination of epigenetic therapies with immunotherapy in ovarian cancer. Immunotherapy has reshaped the therapeutic landscape of cancer, yet its performance in ovarian cancer is disappointing. Most theories aiming to explain this phenomenon revolve around the immunosuppressive TME of ovarian cancer. As such, attempts are made to reverse this situation. In this context 5-azacytidine and HDACi have been found to influence the tumor microenvironment towards the proinflammatory phenotype (i.e., increased infiltration of IFNγ+ T-cells and Natural Killer cells and reduced presence of immunosuppressive M2 macrophages). This shift has been linked towards better responses to immunotherapy as is manifest by the triple combination of DNMTi, HDACi, and anti-PD-1 agents [93,149,150].

## 8. Conclusions and Future Perspectives

Ovarian cancer is the gynecologic malignancy with the highest mortality burden in the Western world. Excluding the recent advent of PARP inhibitors, disease management remains largely unaltered for decades revolving around optimal cytoreduction and cytotoxic chemotherapy. In spite of initial treatment success, most patients will experience recurrence of their disease and eventually succumb. In order to achieve better patient outcomes, further clarification of disease pathophysiology is mandatory. 

In this context, a focus on the epigenetic modifications may help to understand the inner workings of ovarian cancer. Firstly, epigenetic profiling—via high-throughput sequencing of DNA methylation patterns, histone modification signatures, and non-coding RNA expression—has the potential to define molecular subtypes with predictive and prognostic significance. In the coming years, comprehensive epigenetic maps could be incorporated into clinical decision making, guiding treatment choices and improving risk stratification. Secondly, liquid biopsy techniques, especially those focusing on cfDNA methylation, offer promising tools for early detection, disease monitoring, and real-time treatment assessment. However, the clinical translation of such tools will require standardized assays, large-scale validation in diverse populations, and integration with current diagnostic algorithms. The convergence of cfDNA methylation data with AI-driven analytics may further refine diagnostic accuracy and reduce false-positive rates, especially in early-stage disease. Furthermore, RNA methylation and post-transcriptional modifications are increasingly recognized as crucial regulators of gene expression and treatment response. As tools for detecting and modulating RNA modifications improve, new therapeutic targets may be identified within this emerging field. The development of small-molecule inhibitors or RNA-based therapies that target “writer,” “eraser,” or “reader” enzymes may pave the way for the next generation of ovarian cancer treatments.

Available data suggest that gene expression modification is of paramount importance to disease progression and may explain the emergence of treatment resistance. A growing array of clinical trials are already testing DNMTis and HDACis in combination with chemotherapy or TKIs. Despite the fact that initial results are not particularly encouraging, one has to take into account that platinum-resistant disease is largely unresponsive to most available options. Furthermore, the efficacy of epigenetic treatment may be increased as our understanding of the underlying mechanisms expands. There is hope that through manipulation of epigenetic alterations, novel treatment options may be added to the therapeutic arsenal. Moreover, emerging evidence underscores the importance of the immunosuppressive microenvironment in ovarian cancer progression. Epigenetic reprogramming of the TME—through the modulation of CAFs, TAMs, and stromal cells—offers a novel therapeutic axis. Future studies will need to address how epigenetic therapies can be fine-tuned to reshape the TME toward an immune-permissive phenotype, possibly unlocking the full potential of immunotherapy in ovarian cancer. To fully exploit the therapeutic potential of epigenetic interventions, future clinical trials must incorporate biomarker-driven patient selection and dynamic endpoints, such as epigenetic reprogramming or immune landscape modulation. Multi-arm, adaptive trial designs may expedite the evaluation of combinations and help identify patient subgroups that derive the most benefit. This will allow for more precise patient selection and potentially improved outcomes. 

To sum up, ovarian cancer is the most lethal gynecologic malignancy, with limited treatment advances beyond PARP inhibitors. Most patients relapse after initial therapy, highlighting the need for deeper understanding of disease mechanisms. Epigenetic research—focusing on DNA methylation, histone modification, and non-coding RNA—may enable better molecular classification, early detection, and personalized treatment. Liquid biopsies analyzing cfDNA methylation and AI-based analytics could improve diagnosis and disease monitoring. Emerging evidence shows RNA modifications and the TME play major roles in resistance and progression. Epigenetic drugs targeting key enzymes or reprogramming the TME may enhance immunotherapy and treatment response. Ongoing trials with DNMT and HDAC inhibitors show limited results but offer hope when combined with biomarkers and adaptive designs. Expanding epigenetic insights could ultimately lead to more precise, effective, and personalized therapies for ovarian cancer. Therefore, more research is needed so as to increase our understanding of ovarian cancer and thus hopefully pave the way for the development of practices that will help patients in their struggle with this terrible disease.

## Figures and Tables

**Table 2 biomedicines-13-02820-t002:** Clinical trials of DNMTis and HDACis in chemoresistant ovarian cancer.

Author/Year	Therapeutic Agents and Dosing	Study Phase/Participants	Regimen	Outcomes	Notable Findings
Fang et al. (2010) [131]	Decitabine (DNMTi) + carboplatin	Phase I (n:10)	iv Decitabine 10 or 20 mg/m^2^ on d1–5 q28d and iv Carboplatin on d8	CR:1, SD: 3	Satisfactory safety profile
Fu et al. (2011) [132]	Azacitidine (DNMTi) + carboplatin	Phase I-II (n: 29)	sc Azacitidine: 75 mg/m^2^ d1–5 q28d and iv Carboplatin on d2	CR:1, PR:3, SD: 10	PFS> 5months in platinum-resistant patients, Response rate of 22% in platinum-resistant patients
Matei et al. (2012) [130]	Decitabine (DNMTi) + carboplatin	Phase II (n: 17)	iv Decitabine 10 mg/m^2^ on d1–5 q28d and iv Carboplatin on d8	CR:1, PR:5, SD: 6	PFS > 10 months, Response rate of 35%
Glasspool et al. (2014) [133]	Decitabine (DNMTi) + carboplatin	Phase II (n: 29)	sc Decitabine 90 and subsequently 45 mg/m^2^ on d1 q28d and Carboplatin on d8	PR:3, SD: 5	Premature trial termination due to unacceptable toxicity
Zhang et al. (2017) [142]	Decitabine (DNMTi) + carboplatin + paclitaxel	Phase II (n: 40)	iv Decitabine 7 mg/m^2^ on d1-5 q28d and iv Carboplatin and Paclitaxel on d6	CR:1, PR:8, SD: 19	PFS of 8 months and OS of 19 months in platinum-resistant patients, Good safety profile
Matei et al. (2018) [134]	Guadecitabine (DNMTi) + carboplatin	Phase I (n: 20)	sc Guadecitabine 45 to 60 mg/m^2^ (dose escalation) on d1 q28d and iv Carboplatin on d8	PR:3, SD: 6	PFS of 3 months
Oza et al. (2020) [135]	Guadecitabine (DNMTi) + carboplatin	Phase II (n: 100)	sc Guadecitabine 30 mg/m^2^ on d1 q28d and iv Carboplatin on d8	CR + PR: 21	PFS at 6 months: 37%—not statistically significant
Dizon et al. (2012) [138]	Belinostat (HDACi) + carboplatin	Phase II (n: 27)	iv Belinostat 1000 mg/m^2^ d1–5 q21d and iv Carboplatin on d3	CR:1, PR:1, SD: 12	Response rate of 7.4%—premature trial termination due to lack of efficacy—poor safety profile
Matulonis et al. (2015) [139]	Vorinostat (HDACi) + carboplatin + gemcitabine	Phase I (n: 15)	po Vorinostat 200–400 mg (dose-escaltion, once or twice daily) q21d and iv Carboplatin and Gemcitabine on d8	PR:6, SD: 1	Response rate of 40%—premature trial termination due to lack of efficacy—poor safety profile
Burness et al. (2025) [143]	Belinostat (HDACi) + talazoparib	Phase I (n: 25–5 patients with ovarian cancer)	po Talazoparib 0.75–1 mg and Belinostat 500–1000 mg/m^2^	N/a	Favorable safety profile

## Data Availability

No new data were created or analyzed in this study.

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
