# Peer review of "Epigenetics in Ovarian Cancer: A Review of Current Knowledge and Future Perspectives"

_biomedicines, 2025, doi:10.3390/biomedicines13112820_

Round 1

Reviewer 1 Report

Comments and Suggestions for Authors

Although the chosen topic is current and of great interest, this review requires extensive revision and completion of missing information.

Here are some observations that could help improve the manuscript:

It is not clear what the purpose of this manuscript is.

What are the objectives of the review?

The epigenetic mechanisms responsible for the occurrence and evolution of ovarian cancer are not clearly described. Also, the utility of epigenetic factors in the clinic as biomarkers and their potential as therapeutic targets should be emphasized.

It would be desirable to emphasize the role of epigenetic therapy in ovarian cancer and also mention clinical trials, if any, and the challenges generated.

All this information could be integrated into tables and figures to support the information presented.

There is no chapter describing future perspectives.

I regret that the manuscript cannot be published in the presented form in the journal Biomedicines, which has a Journal Rank of JCR—Q1 and an Impact Factor of 3.9.

Comments on the Quality of English Language

 The English could be improved to more clearly express the research.

Author Response

Comment 1: What are the objectives of the review?

Response 1: We thank the Reviewer for the comment. A paragraph has been added in order to further clarify the aims of the review. Please refer to lines 67-74 “This review endeavors to highlight the role of epigenetics in all aspects of ovarian cancer. The initial aim is to investigate epigenetic modification in disease natural his-tory and its pathophysiology. Knowledge of the underlying molecular processes may help develop practice-changing clinical applications in disease screening and diagnosis. Moreover, epigenetic manipulation via a series of therapeutic agents has already yielded results in other fields of medicine. In this context, there is sufficient evidence to speculate that such drugs may be efficacious in ovarian cancer as well. As such, this review may offer insight into future perspectives of ovarian cancer management.”

Comment 2: The epigenetic mechanisms responsible for the occurrence and evolution of ovarian cancer are not clearly described. Also, the utility of epigenetic factors in the clinic as biomarkers and their potential as therapeutic targets should be emphasized. It would be desirable to emphasize the role of epigenetic therapy in ovarian cancer and also mention clinical trials, if any, and the challenges generated. All this information could be integrated into tables and figures to support the information presented.

Response 2: We thank the Reviewer  for the comment. We have now included in the manuscript two new Tables to support the presented information.

Table 1 (line 163) may help clarify the influence of epigenetic mechanisms on ovarian cancer pathogenesis and also highlight potential novel clinical applications that would be helpful in ovarian cancer management.

Table 2 (line 321) presents the most notable clinical trials of DNMTis and HDACis in chemoresistant ovarian cancer. On this table the reader may find key characteristics of each trial like the number of participants or the regimen used, as well as other important information regarding response rates, clinical efficacy and toxicities.

Comment 3:  There is no chapter describing future perspectives.

Response 3: We thank the Reviewer for the insight. In this context, the conclusion’s section was extensively expanded and modified to include future perspectives

Line 373-389: epigenetic profiling—via high-throughput sequencing of DNA methylation patterns, histone modification signatures, and non-coding RNA expression—has the potential to define molecular subtypes with predictive and prognostic significance. In the coming years, comprehensive epigenetic maps could be incorporated into clinical decision-making, guiding treatment choices and improving risk stratification. Secondly, liquid biopsy techniques, especially those focusing on cfDNA methylation, offer promising tools for early detection, disease monitoring, and real-time treatment assessment. However, the clinical translation of such tools will require standardized assays, large-scale validation in diverse populations, and integration with current diagnostic algorithms. The convergence of cfDNA methylation data with AI-driven analytics may further refine diagnostic accuracy and reduce false-positive rates, especially in ear-ly-stage disease. Furthermore, RNA methylation and post-transcriptional modifications are increasingly recognized as crucial regulators of gene expression and treatment response. As tools for detecting and modulating RNA modifications improve, new therapeutic targets may be identified within this emerging field. The development of small-molecule inhibitors or RNA-based therapies that target "writer," "eraser," or "reader" enzymes may pave the way for the next generation of ovarian cancer treatments.

Line 392-396: A growing array of clinical trials are already testing DNMTis and HDACis in combination with chemotherapy or TKIs. Despite the fact that initial results are not particularly encouraging, one has to take into account that platinum-resistant disease is largely unresponsive to most available options. Furthermore, the efficacy of epigenetic treatment may be increased as our understanding of the underlying mechanisms expands.

Line 398-409: emerging evidence underscores the importance of the immunosuppressive microenvironment in ovarian cancer progression. Epigenetic reprogramming of the TME—through the modulation of CAFs, TAMs, and stromal cells—offers a novel therapeutic axis. Future studies will need to address how epigenetic therapies can be fine-tuned to reshape the TME toward an immune-permissive phenotype, possibly un-locking the full potential of immunotherapy in ovarian cancer. To fully exploit the therapeutic potential of epigenetic interventions, future clinical trials must incorporate biomarker-driven patient selection and dynamic endpoints, such as epigenetic reprogramming or immune landscape modulation. Multi-arm, adaptive trial designs may expedite the evaluation of combinations and help identify patient subgroups that de-rive the most benefit. This will allow for more precise patient selection and potentially improved outcomes.

Reviewer 2 Report

Comments and Suggestions for Authors

This paper tackles a highly relevant and timely topic in the field of gynecologic oncology. Ovarian cancer remains one of the deadliest cancers affecting women, largely due to its late-stage diagnosis and the slow progress in developing effective treatments. By focusing on epigenetics ad a rapidly advancing area in cancer biology the review highlights its clinical importance. It examines how epigenetic mechanisms contribute to ovarian cancer development and resistance to therapy, aligning with ongoing efforts to personalize treatment and improve early detection. The review provides a well-timed and original synthesis of current knowledge and future directions for epigenetics in ovarian cancer, combining fundamental biological insights with clinical applications. This makes it a valuable resource for both researchers and clinicians aiming to advance therapeutic innovation.

One of the paper’s key strengths is its thorough overview of the multiple layers of epigenetic regulation, including DNA methylation, histone modifications, non-coding RNAs, chromatin remodeling, and RNA methylation. Each mechanism is clearly explained and linked to ovarian cancer progression and treatment challenges. The paper also appropriately emphasizes the importance of developing subtype-specific biomarkers and personalized treatment strategies.

The discussion of liquid biopsy and circulating tumor DNA methylation as emerging diagnostic tools reflects the increasing clinical interest in non-invasive methods for early detection and monitoring. Moreover, the paper explores the therapeutic potential of targeting epigenetic regulators, offering a forward-looking perspective. These targeted approaches are particularly promising for overcoming chemoresistance and improving immunotherapy outcomes, which represent two major hurdles in ovarian cancer care. Grounded firmly in current scientific literature, the review also identifies gaps in clinical translation, enhancing its credibility and usefulness for both researchers and clinicians striving to bridge basic research and patient care.

Overall, the manuscript is comprehensive and demonstrates a strong grasp of the topic, drawing on a broad range of studies to support its arguments. Some minor issues include occasional abrupt transitions and moments of overly technical explanation. The manuscript would benefit from

The conclusion effectively summarizes the main points but tends to be somewhat broad and cautious, which may weaken its impact. A clearer emphasis on the unique contributions of epigenetics to understanding and treating ovarian cancer, along with highlighting the most promising diagnostic and therapeutic strategies, would strengthen the closing remarks.

Author Response

Comment 1: Some minor issues include occasional abrupt transitions and moments of overly technical explanation.

Response 1: We thank the Reviewer for his/her comment. We have now revised the manuscript so as to facilitate the conveyance of message. This includes the incorporation of two tables. Table 1 (line 163) may help clarify the influence of epigenetic mechanisms on ovarian cancer pathogenesis and also highlight potential novel clinical applications that would be helpful in ovarian cancer management. Table 2 (line 321) presents the most notable clinical trials of DNMTis and HDACis in chemoresistant ovarian cancer. On this table the reader may find key characteristics of each trial like the number of participants or the regimen used, as well as other important information regarding response rates, clinical efficacy and toxicities.

Comment 2: The conclusion effectively summarizes the main points but tends to be somewhat broad and cautious, which may weaken its impact. A clearer emphasis on the unique contributions of epigenetics to understanding and treating ovarian cancer, along with highlighting the most promising diagnostic and therapeutic strategies, would strengthen the closing remarks.

Response 2: We thank Reviewer 2 for the comment. In this context, the conclusion’s section was extensively expanded and modified to include future perspectives

Line 373-389: epigenetic profiling—via high-throughput sequencing of DNA methylation patterns, histone modification signatures, and non-coding RNA expression—has the potential to define molecular subtypes with predictive and prognostic significance. In the coming years, comprehensive epigenetic maps could be incorporated into clinical deci-sion-making, guiding treatment choices and improving risk stratification. Secondly, liquid biopsy techniques, especially those focusing on cfDNA methylation, offer prom-ising tools for early detection, disease monitoring, and real-time treatment assessment. However, the clinical translation of such tools will require standardized assays, large-scale validation in diverse populations, and integration with current diagnostic algorithms. The convergence of cfDNA methylation data with AI-driven analytics may further refine diagnostic accuracy and reduce false-positive rates, especially in ear-ly-stage disease. Furthermore, RNA methylation and post-transcriptional modifica-tions are increasingly recognized as crucial regulators of gene expression and treat-ment response. As tools for detecting and modulating RNA modifications improve, new therapeutic targets may be identified within this emerging field. The development of small-molecule inhibitors or RNA-based therapies that target "writer," "eraser," or "reader" enzymes may pave the way for the next generation of ovarian cancer treat-ments.

Line 392-396: A growing array of clinical trials are already testing DNMTis and HDACis in combi-nation with chemotherapy or TKIs. Despite the fact that initial results are not particularly encouraging, one has to take into account that platinum-resistant disease is largely unresponsive to most available options. Furthermore, the efficacy of epigenetic treatment may be increased as our understanding of the underlying mechanisms expands.

Line 398-409: emerging evidence underscores the importance of the immunosuppressive microenvi-ronment in ovarian cancer progression. Epigenetic reprogramming of the TME—through the modulation of CAFs, TAMs, and stromal cells—offers a novel therapeutic axis. Future studies will need to address how epigenetic therapies can be fine-tuned to reshape the TME toward an immune-permissive phenotype, possibly un-locking the full potential of immunotherapy in ovarian cancer. To fully exploit the therapeutic potential of epigenetic interventions, future clinical trials must incorporate biomarker-driven patient selection and dynamic endpoints, such as epigenetic repro-gramming or immune landscape modulation. Multi-arm, adaptive trial designs may expedite the evaluation of combinations and help identify patient subgroups that de-rive the most benefit. This will allow for more precise patient selection and potentially improved outcomes.

Round 2

Reviewer 1 Report

Comments and Suggestions for Authors

The authors have selected a timely and important topic, as ovarian cancer is the gynecological malignancy with the highest mortality rate globally.

The current version of the article shows significant improvement; however, I would like to emphasize the need for additional modifications, which I have marked in red in the manuscript.

  1. L47-53: The paragraph is quite lengthy. Please revise it for better clarity.
  2. Tables 1 and 2  should be referenced in the text. L166-172: The bullet point is unnecessary.
  3. What is INO80? Please provide a definition.
  4. L207-209: Could you clarify your statement? How can "epitranscriptomics" contribute to understanding the molecular basis of ovarian cancer?
  5. L210-231: This section needs restructuring. Please divide it into three subchapters to clearly present the following topics: DNA modifications, histone modifications, and non-coding RNA modifications (including miRNA, lncRNA, and circRNA), all in relation to the tumor microenvironment (TME). Additionally, include a discussion on emerging epigenetic therapies that stem from TME reprogramming in ovarian cancer.
  6. The conclusion of this study is unclear and needs to be improved.

Overall, the article could be a substantial contribution to the journal. Therefore, I recommend the manuscript for publication after the authors have considered major changes and updates.

Comments on the Quality of English Language

The English could be improved to more clearly express the research.

Author Response

Comment: L47-53: The paragraph is quite lengthy. Please revise it for better clarity

Response: We thank the reviewer for the comment. The paragraph has been rewritten in order to enhance comprehension (L 53-59 – track changes enabled - This large group is further subdivided into high grade serous ovarian carcinomas (HGSOC), low grade serous ovarian carcinomas (LGSOC), endometrioid carcinomas (EC) and clear cell carcinomas (CCC). Both HGSOC and LGSOC originate from the fi-brial or ovarian surface epithelium, but their mutational status is quite different as the former is characterized by p53 and BRCA mutations, while the latter harbors muta-tions of KRAS and BRAF. Moreover, EC and CCC may be associated with endometrio-sis and mutations of PTEN and ARID1 )

Comment: Tables 1 and 2 should be referenced in the text

Response: We thank the reviewer for the comment. Both tables are now mentioned in the text (TABLE 1 at L 85-86 – track changes enabled: Some of the most prominent genes or pathways subjected to epigenetic modification during ovarian cancer pathogenesis are listed on TABLE 1. and TABLE 2 at L 494-495 – track changes enabled: Clinical trials of DNMTis and HDACis in chemoresistant ovarian cancer are shortly displayed on TABLE 2.).

Comment: L166-172: The bullet point is unnecessary. What is INO80? Please provide a definition.

Response: We thank the Reviewer for the comment. The bullets have been incorporated in the text and a short description for the chromatin regulator INO80 has been added at L 175-178  - track changes enabled:  nucleosome remodelers like the a series of enzyme complexes, among the most noted are the ATP-dependent chromatin remodeling complexes Chromodomain helicase DNA binding (CHD), Inositol requiring 80 (INO80), Imitation switch (ISWI) and Switch/sucrose non-fermentable (SWI/SNF, also referred to as BAF)

Comment: L207-209: Could you clarify your statement? How can "epitranscriptomics" contribute to understanding the molecular basis of ovarian cancer?

Response: We thank the Reviewer for the comment. A short clarification has been added (L 220-222 track changes enabled: as the mapping of RNA modifications transcriptome-wide could enhance our under-standing of its pathogenesis. In this context, novel biomarkers may be developed and manipulation of epigenetic alterations could lead to new treatment options)

Comment: L210-231: This section needs restructuring. Please divide it into three subchapters to clearly present the following topics: DNA modifications, histone modifications, and non-coding RNA modifications (including miRNA, lncRNA, and circRNA), all in relation to the tumor microenvironment (TME). Additionally, include a discussion on emerging epigenetic therapies that stem from TME reprogramming in ovarian cancer.

Response: We thank the Reviewer for the comment. This section of the text has been extensively rewritten as per the instructions of the Reviewer (L 228-361 track changes enabled:  As it may be expected, epigenetic modification is central to the formation and the dy-namics of the TME. Modern understanding of cancer biology has highlights the TME as crucial to disease physical history and response to treatment. Mounting evidence sug-gest that TME characteristics actively direct disease course

Evidence from studies in various malignancies demonstrates that stromal fibro-blasts can acquire DNA methylation profiles similar to adjacent malignant cells, indi-cating reciprocal epigenetic crosstalk between the stroma and tumor compartments. In this context, cancer-associated fibroblasts (CAFs) secrete soluble factors that modulate the methylation status of epithelial ovarian cancer cells, promoting widespread tran-scriptional changes, EMT, and acquisition of stem-like features—effects that can be amenable to therapeutic targeting. A key mediator of this interaction is TGF-β1, which is released by fibroblasts in response to tumor-derived signals. In ovarian cancer, TGF-β not only induces global DNA hypermethylation and EMT through upregulation of DNA methylation, but also contributes to the reprogramming of fibroblasts into pro-tumorigenic CAFs that secrete prometastatic mediators. This establishes a self-sustaining feedback loop wherein fibroblast–tumor interactions perpetuate epige-netic remodeling, CAF activation, and tumor invasion. Targeting this loop through the combined use of demethylating agents and TGF-β pathway inhibitors therefore repre-sents a promising therapeutic avenue to disrupt the epigenetic reinforcement of the ovarian cancer microenvironment[88]. Moreover, DNA methylation plays a pivotal role in shaping the immunosuppressive landscape, particularly through its impact on myeloid-derived suppressor cells (MDSCs). These heterogeneous immune cells expand in response to malignant transformation and inflammation, where they suppress an-ti-tumor T-cell activity and promote tumor progression. Elevated MDSC levels in ovarian cancer patients are strongly correlated with reduced overall survival, high-lighting their clinical relevance. Pro-tumorigenic cytokines such as interleukin-6 (IL-6) and interleukin-10 (IL-10) enhance MDSC recruitment and function through activa-tion of the STAT3 signaling pathway, while VEGF and adenosine secreted by ovarian cancer cells further facilitate their accumulation and immune-suppressive effects. Im-portantly, recent evidence indicates that MDSCs undergo DNA methyltransferase 3A- and prostaglandin E2-dependent hypermethylation, a process essential for their acqui-sition of immunosuppressive properties. This epigenetic reprogramming not only re-inforces immune evasion within the TME but also underscores the potential of target-ing methylation pathways as a novel therapeutic strategy to restore anti-tumor im-munity in ovarian cancer[89,90].

In addition to DNA methylation, histone modifications within the ovarian cancer TME have emerged as key regulators of tumor progression, metastasis, and therapeutic response by influencing both stromal and immune cell behavior. Beyond cancer cells, epigenetic alterations—including histone methylation and acetylation—profoundly shape the phenotype of stromal components such as fibroblasts, adipocytes, and im-mune cells. Notably, nicotinamide N-methyltransferase (NNMT) was found to be up-regulated in CAFs within metastatic ovarian cancer lesions, where it depletes S-adenosyl methionine, the principal methyl donor for histone methylation. This de-pletion leads to reduced global histone methylation, promoting transcriptional repro-gramming that enhances collagen secretion, cytokine release, and metastatic poten-tial[91]. Conversely, inhibition of NNMT restores histone methylation and suppresses expression of metastasis-associated genes, underscoring its central role in stromal re-modeling. Platinum-based chemotherapy further alters histone dynamics in the TME by inducing IL-6 secretion from CAFs, which drives stemness-associated pathways in residual tumor cells. Additionally, epigenetic silencing of immune-related cytokines through DNA and H3K27 methylation limits immune infiltration, contributing to the “cold” immune phenotype characteristic of OC. Restoration of histone methylation balance using hypomethylating or histone methyltransferase inhibiting agents can re-activate immune signaling pathways, including CCL5–CXCL9/CXCL10 axes, thereby enhancing T-cell infiltration and responsiveness to immune checkpoint blockade[92]. Furthermore, cytokines and signaling molecules such as IL-33, IL-6, GM-CSF, IL-4, and IL-10 orchestrate histone modification patterns in tumor-infiltrating MDSCs and tu-mor-associated macrophages (TAMs), enhancing their immunosuppressive functions. For example, IL-33 promotes survival of MDSCs by increasing activating histone marks like H3K4me3 and H3K14ac, while IL-6 and GM-CSF induce STAT3- and C/EBPβ-dependent transcriptional programs that regulate H3K4me3 enrichment at promoters of immunosuppressive genes such as Arg1, NOS2, and COX2. Similarly, IL-4-driven STAT6 signaling upregulates JMJD3, a histone demethylase that removes repressive H3K27me3 marks from TAM-related genes, sustaining the tu-mor-promoting macrophage phenotype. IL-10 also contributes to this suppressive mi-lieu by recruiting histone demethylases such as KDM6A to remove H3K27me3 and promote transcription of Hotairm1, facilitating nuclear localization of S100A9 and expansion of MDSCs. Collectively, these findings highlight histone modification as a central epigenetic mechanism coordinating tumor–stroma–immune interactions in ovarian cancer and suggest that targeting histone-modifying enzymes offers a prom-ising strategy to reprogram the TME and improve therapeutic outcomes[93,94].

Beyond DNA methylation and histone modifications, ncRNAsare emerging as key instrumens of epigenetic and post-translational modification. Non-coding RNAs, in-cluding microRNAs (miRNAs), long non-coding RNAs (lncRNAs), and circular RNAs (circRNAs), play crucial regulatory roles in the TME, influencing cancer progression, metastasis, immune evasion, and therapy resistance[94]. Genetic alterations—both germline and somatic—can occur within ncRNA genes, affecting their expression and function. For example, differential miRNA expression patterns have been explored as diagnostic biomarkers for hereditary breast and ovarian cancers, while somatic mutations in miRNA genes and their target sites (such as in 3′UTRs) can disrupt regulatory interactions and promote oncogenesis[95]. Within the TME, communication among tumor, stromal, and immune cells is mediated by cytokines and ncRNAs. Cytokines such as IL-6, IL-10, and IFN-γ can modulate ncRNA expression, which in turn regulates pathways like PI3K/Akt, NF-κB, and JAK/STAT[96]. For instance, lncRNA ZFAS1 promotes cancer progression by sponging miRNA-6499-3p and upregulating CCL5, while lncRNA INCR1 enhances PD-L1 expression through IFN-γ signaling. Exosomes act as key ncRNA carriers within the TME, transporting miRNAs, lncRNAs, and circRNAs that regulate tumor growth, angiogenesis, immune suppression, and drug resistance. Exosomal ncRNAs such as miRNA-1246 and miRNA-6126 modulate path-ways involved in ovarian cancer progression and therapy response[97]. These vesicles also influence immune cells—promoting macrophage polarization, impairing T-cell function, and altering dendritic cell activity—thus shaping immune evasion. Cancer-associated fibroblasts, activated by growth factors and hypoxia, express ncRNAs that remodel the extracellular matrix and enhance metastasis. In ovarian cancer, CAF-derived CXCL14 increases lncRNA LINC00092, promoting invasion[98]. Similarly, TAMs, primarily of the M2 phenotype, rely on ncRNAs for recruitment and polarization; for example, miRNA-149 suppresses TAM infiltration, while siRNAs tar-geting VEGF and PIGF reprogram TAMs to an antitumor M1 state[99]. Non coding RNAs also regulate T-cell and B-cell functions in the TME. Modulation of immune checkpoints (PD-1/PD-L1) via siRNAs can restore T-cell cytotoxicity, while ZFP91 knockdown enhances T-cell metabolism and activity[100]. B-cell activity is influenced by ncRNAs downstream of CXCL13/CXCR5 signaling, affecting proliferation and migration. Finally, ncRNAs govern EMT, a key process in metastasis and drug resistance. Downregulation of miRNA-200b/c promotes EMT and tamoxifen resistance, while re-storing their expression reverses these effects[101]. Collectively, ncRNAs serve as both mediators and messengers within the TME—regulating cytokine signaling, intercellu-lar communication, immune responses, and treatment sensitivity—making them promising diagnostic biomarkers and therapeutic targets in cancer.

With regard to the far-reaching influence of epigenetics on the TME, it would be rational to consider the therapeutic potential of epigenetic modulation with the aim to reprogram immune and stromal components of the ovarian cancer TME toward anti-tumor activity. DNA methylation inhibitors, such as 5-azacytidine (5-aza) and 5-aza-2’-deoxycytidine (5-aza-dC), can reprogram immunosuppressive M2 TAMs into proinflammatory M1 phenotypes, enhancing tumor sensitivity to chemotherapy such as paclitaxel[102]. Moreover, combination therapies involving DNA methylation modulating agents and histone deacetylase inhibiting agents (e.g., trichostatin A, TSA) further remodel the cytokine milieu, promoting antitumor immune responses[103]. In regulatory T cells (Tregs), modulation of FOXP3 methylation alters their suppressive capacity, with hypomethylating agents transiently increasing FOXP3 expression but ultimately reducing Treg-mediated immunosuppression, thereby enhancing check-point inhibitor efficacy[103]. Histone modifications also represent a promising thera-peutic avenue. Selective inhibition of specific HDACs (e.g., HDAC6 by ACY-1215, HDAC3 by RGFP966) enhances CD8⁺ T-cell cytotoxicity while limiting Treg function, whereas inhibitors of EZH2 (UNC1999, EPZ005687, CPI-1205) promote NK cell activation and facilitate conversion of Tregs to Th1-like cells, boosting IFNγ production[104]. Targeting acetyltransferases (EP300/CBP inhibitors) similarly dis-rupts FOXP3 acetylation, reducing Treg stability and improving immune activation within the OC TME. RNA modification inhibitors have recently emerged as an additional layer of epigenetic intervention. Inhibitors of m⁶A regulators such as FTO (e.g., Dac51) and ALKBH5 (ALK-04) suppress the infiltration of immunosuppressive MDSCs and Tregs, enhancing anti-PD-1 therapy efficacy[105]. Similarly, METTL3 inhibitors (STM2457) and pseudouridine synthase (PUS7) inhibitors (C17) demonstrate anti-tumor activity and improved survival in preclinical models[106]. Collectively, these findings underscore the therapeutic promise of combining epigenetic drugs with immunotherapies to reprogram the ovarian cancer TME. By targeting DNA methylation, histone modification, and RNA modification pathways, it may be possible to overcome immune resistance, restore effector cell activity, and enhance the efficacy of immune checkpoint blockade in ovarian cancer.)

Comment: The conclusion of this study is unclear and needs to be improved

Response: We thank the Reviewer for the comment. A paragraph has been added to summarize the conclusions and future perspectives. (L 560-572 track changes enabled: To sum up, ovarian cancer is the most lethal gynecologic malignancy, with limited treatment advances beyond PARP inhibitors. Most patients relapse after initial therapy, highlighting the need for deeper understanding of disease mechanisms. Epigenetic research—focusing on DNA methylation, histone modification, and non-coding RNA—may enable better molecular classification, early detection, and personalized treatment. Liquid biopsies analyzing cfDNA methylation and AI-based analytics could improve diagnosis and disease monitoring. Emerging evidence shows RNA modifications and the TME play major roles in resistance and progression. Epigenetic drugs targeting key enzymes or reprogramming the TME may enhance immunotherapy and treatment response. Ongoing trials with DNMT and HDAC inhibitors show limited results but offer hope when combined with biomarkers and adaptive designs. Expanding epigenetic insights could ultimately lead to more precise, effective, and personalized therapies for ovarian cancer.)

Round 3

Reviewer 1 Report

Comments and Suggestions for Authors

The authors have made the requested changes and additions.